# Transmission-Reflection-Integrated Multifunctional Passive Metasurface for Entire-Space Electromagnetic Wave Manipulation

**DOI:** 10.3390/ma16124242

**Published:** 2023-06-08

**Authors:** Shunlan Zhang, Weiping Cao, Tiesheng Wu, Jiao Wang, Heng Li, Yanliang Duan, Haoyu Rong, Yulong Zhang

**Affiliations:** 1Guangxi Key Laboratory of Wireless Wideband Communication and Signal Processing, Guilin 541004, China; 2Key Laboratory of Cognitive Radio and Information Processing (Ministry of Education), Guilin University of Electronic Technology, Guilin 541004, China; 3School of Information and Communication, Guilin University of Electronic Technology, Guilin 541004, China

**Keywords:** metasurface, multifunctional, passive, entire space

## Abstract

In recent years, many intriguing electromagnetic (EM) phenomena have come into being utilizing metasurfaces (MSs). However, most of them operate in either transmission or reflection mode, leaving the other half of the EM space completely unmodulated. Here, a kind of transmission-reflection-integrated multifunctional passive MS is proposed for entire-space electromagnetic wave manipulation, which can transmit the x-polarized EM wave and reflect the y-polarized EM wave from the upper and lower space, respectively. By introducing an H-shaped chiral grating-like micro-structure and open square patches into the unit, the MS acts not only as an efficient converter of linear-to-left-hand circular (LP-to-LHCP), linear-to-orthogonal (LP-to-XP), and linear-to-right-hand circular (LP-to-RHCP) polarization within the frequency bands of 3.05–3.25, 3.45–3.8, and 6.45–6.85 GHz, respectively, under the x-polarized EM wave, but also as an artificial magnetic conductor (AMC) within the frequency band of 12.6–13.5 GHz under the y-polarized EM wave. Additionally, the LP-to-XP polarization conversion ratio (PCR) is up to −0.52 dB at 3.8 GHz. To discuss the multiple functions of the elements to manipulate EM waves, the MS operating in transmission and reflection modes is designed and simulated. Furthermore, the proposed multifunctional passive MS is fabricated and experimentally measured. Both measured and simulated results confirm the prominent properties of the proposed MS, which validates the design’s viability. This design offers an efficient way to achieve multifunctional meta-devices, which may have latent applications in modern integrated systems.

## 1. Introduction

Metamaterials have rapidly developed during the past few decades due to their exceptional electromagnetic (EM) properties, which are unavailable in nature. In general, metamaterials, which are often made up of artificially periodic or quasiperiodic structures with sub-wavelength scales, generate many brilliant phenomena and remarkable devices [1,2,3,4,5,6,7,8]. By elaborately designing subwavelength meta-structures across the interface, many significant applications based on metasurfaces (MSs) were made possible, including beam scanning [9,10], focusing [11,12], vortex-beam realization [13,14], polarization conversion [15,16,17], holographic imaging [18,19], and anomalous scattering [20,21]. Since Cui et al. introduced the idea of encoding MSs using binary codes [22], the number of coded MSs has been growing quickly [23]. Based on this concept, digital MSs and programmable MSs were realized, which can dynamically control the EM waves by using PIN diodes and field-programmable gate arrays [24]. Many noteworthy accomplishments have been acquired, but the majority of MSs up to now cannot meet the requirements of modern integrated systems due to their simplistic functions. To meet the rapid development of modern devices and systems, MSs are generally required to integrate various functions by utilizing different attributes. Active MSs, including programmable, tunable, and reconfigurable MSs, have been reported and can realize multiple functionalities [25,26]. These different functions realized by active devices can be switched by various controlling signals, resulting in multifunctionalities. However, the design complexity of active MSs greatly limits their applications. 

In recent years, a lot of bifunctional or multifunctional MSs by altering frequencies, helicity, and polarization have been investigated on passive MSs [27,28]. However, the majority of multifunctional MSs that have been reported so far only work in either transmission or reflection modes, leaving the other space unutilized. The benefits of variable EM responses and multifunctional wave front control make entire-space EM wave manipulation a hot topic [29,30,31,32]. For instance, the entire-space multifunctional control of EM waves with active MSs has also been accomplished using reconfiguration techniques [30,31,32]. However, complicated biasing networks are required for the design of active MSs, which inevitably raises system costs and loss. Additionally, some multifunctional passive MSs for the entire-space modulation of EM waves have been described in refs. [33,34,35,36], where the two mainstreams are helicity-dependent metasurfaces [33,34] and anisotropic metasurfaces [35,36]. Different functionality can be achieved in a single device by varying the polarization and helicity of incident waves. However, it is crucial to remember that all of the aforementioned studies concentrated on bifunctional realizations, whereas more research needs to be performed on multifunctional passive MSs.

In this study, in order to meet the simple and multifunctional requirements of modern miniaturized and highly integrated systems, theory and experiment are combined to show a transmission-reflection-integrated multifunctional passive MS that integrates four distinct functionalities into a single MS, including the conversion of the linear-to-left-hand circular (LP-to-LHCP), linear-to-orthogonal (LP-to-XP), and linear-to-right-hand circular (LP-to-RHCP) polarization in the frequency bands of 3.05–3.25, 3.45–3.8, and 6.45–6.85 GHz, respectively, under the x-polarized wave from the upper half-space, as well as an artificial magnetic conductor (AMC) function in the frequency band of 12.6–13.5 GHz under the y-polarized wave from the lower half-space, as shown in Table 1. Moreover, the polarization conversion ratio (PCR) is up to −0.52 dB at 3.8 GHz. The presented MS unit cell consists of four precisely designed metal patterns divided by three substrate layers. For transmitted waves, the top three metal patterns can function as a polarization converter. Additionally, metal patterns 4 and 3 work as an artificial magnetic conductor for reflected waves. Furthermore, all of these transmission functions are unaffected under large oblique incidence angles up to 45°. In the meantime, low scattering and in-phase reflection properties can be achieved in the reflection mode. The schematic diagram is presented in Figure 1, which shows the full-space radiation’s principle of operation. To validate the multifunctional performance, a 26 × 26 array is simulated, designed and fabricated. We use experiments to show that the suggested MS is capable of realizing four different functionalities. In contrast to earlier research, our ideas provide a feasible method to realize multifunctional passive MS operating throughout the space, which can result in a lot of intriguing applications in different spaces and frequency bands. The structure of this essay is as follows. Section 2 depicts the MS design, performance, physical mechanism, and simulated results. Measured results are given in Section 3. Additionally, Section 4 draws the main conclusions.

## 2. MSs Design and Analysis

### 2.1. Ms Unit Cell Design 

The suggested unit cell’s structure is described in Figure 2, with an element periodicity of 18 mm. There are three substrate layers and four metal layers in each unit. On the Arlon AD255A (tm) substrate with εr = 2.55, the metal patterns 1, 2, and 3 are etched. Additionally, on the substrate F4B with εr = 2.65, the metal layer 4 is etched. The orthogonal metal grating of the metal layers 1 and 3 is to enhance the conversion properties of the x-polarized EM wave. The metal layer 2 is a metal-H, which is canted 45°, and acts as the polarization converter. Four open square patches in metal layer 4 serve as a man-made magnetic conductor combined with the metal grating of metal layer 3. The optimized parameters for the unit marked in Figure 2 are as follows: px = py = 18 mm, g = 4.05 mm, w = 1.35 mm, g1 = 0.33 mm, x1 = 5.5 mm, y1 = 1 mm, h1 = 3.175 mm, h2 = 1.524 mm, h3 = 1.2 mm, dl1 = dl2 = 9 mm, and dw = 2.7 mm.

### 2.2. Performance Analysis

To look into the multifunctional performances of the suggested MS, we used Ansoft HFSS to accomplish simulations. Periodic boundary conditions and two Floquet ports were adopted to model an infinite array. Simulation results of the MS operating in the transmission and reflection modes were performed. For the transmission mode, the incident wave is the x-polarized EM wave from the upper half-space. Additionally, for the reflection mode, the incident wave is the y-polarized EM wave from the lower half-space.

#### 2.2.1. Transmission Performance

Transmission coefficients of the x-polarization (co-polarization) and the y-polarization (cross-polarization) wave are defined as *t_xx_* =|Ext|/|Exi| and *t_yx_* =|Eyt|/|Exi|, respectively, where Eyt is the transmitted electric field amplitudes along the *y*-axis, whereas Ext and Exi are the transmitted and incident electric field amplitudes along the *x*-axis, respectively, so the polarization conversion rate (PCR) can be determined by [37]
(1)$$\mathrm{PCR}\;={\left|t_{yx}\right|}^{2}/\left({\left|t_{yx}\right|}^{2}+{\left|t_{xx}\right|}^{2}\right)$$

The simulation results of the MS working in the transmission mode are shown in Figure 3, which describes the simulated transmission coefficients, phase differences, PCRs, reflection coefficients, refractive indexes, and different transmission coefficients with different periods p. It is evident from Figure 3a that conversions of LP-to-LHCP, LP-to-RHCP, and LP-to-XP (the cross components of the transmission coefficient tyx are taken above −3 dB and are higher by 6 dB than co-components txx over the required frequency band) can be obtained within the frequency bands 3.05–3.25, 6.45–6.85, and 3.45–3.8 GHz, respectively. From Figure 3b, we can find that the PCRs are high in the range of 3.45–3.8 GHz, and the PCR can be up to −0.52 dB at 3.8 GHz. This shows that the suggested MS has good LP-to-XP conversion capability. At the same time, it can be found that the proposed MS possesses a weak resonant peak at 3.65 GHz and a strong resonant peak at 6.42 GHz in transmission mode, as described in Figure 3b. Figure 3c describes the extractive refractive indexes of the proposed MS based on Kramers–Kronig relationship. From Figure 3c, we can find that the refractive indexes are negative in the range of 3.58–5, 5.3–5.5, and 6.7–7.5 GHz. This shows that the suggested MS has many novel electromagnetic properties, such as negative refraction, the reversed Doppler effect, and reversed Cerenkov radiation. It is clear from Figure 3d that the conversion performance of the proposed MS from the x-polarized EM wave to the y-polarized EM wave deteriorates with the increase in periods p; however, the bandwidth will be narrow if p is small.

To analyze the influence of oblique incidence on the transmission performance, we performed simulations under different incident angles θ. Figure 4 presents the transmission coefficients and phase differences under different incident angles θ. It can be found that the effect of oblique incidence angle on transmission coefficients and phase differences is no obvious mismatch up to the oblique incidence angle of 45°, which indicates that the proposed MS possesses an angular stability of the response under the transmission mode.

We further discuss the mechanism behind polarization conversion under the transmission mode. Supposing that the plane wave with the x-direction electric field is perpendicularly incident on the MS, we resolve it into two eigenmodes along the u-axis and the v-axis, as depicted in Figure 5. The electric fields of an incident electromagnetic wave and electromagnetic waves that are transmitted can be expressed as
(2)$${\vec{E}}_{i}=E_{0}{\hat{e}}_{x}=E_{ui}e^{j\varphi }{\hat{e}}_{u}+E_{vi}e^{j\varphi }{\hat{e}}_{v}.$$
(3)$${\vec{E}}_{t}=t_{uu}E_{ui}e^{j(\varphi }{}^{+\varphi_{u})}{\hat{e}}_{u}+t_{vv}E_{vi}e^{j(\varphi }{}^{+\varphi_{v})}{\hat{e}}_{v}^{}=E_{ut}e^{j(\varphi }{}^{+\varphi_{u})}{\hat{e}}_{u}+E_{vt}e^{j(\varphi }{}^{+\varphi_{v})}{\hat{e}}_{v}^{}.$$
where *E_ui_* and *E_vi_* are the incident electric fields along the u-axis and v-axis, respectively, *t_uu_* and *t_vv_* are the transmission coefficients along the u-axis and v-axis, respectively, and *φ_u_* and *φ_v_* are the phases along the u-axis and v-axis, respectively. Owing to the presented structure’s anisotropy, there are phase differences Δ*φ_vu_* between *φ_u_* and *φ_v_*. The presented structure works in the transmission mode, and when the conditions *t_uu_* ≈ *t_vv_* and Δ*φ_vu_* ≈ ±90° are satisfied simultaneously, then the circular polarization conversion can be obtained. If Δ*φ_vu_* ≈ 90°, the conversion of LP-to-LHCP will be achieved within the frequency band 3.05–3.25 GHz, as shown in Figure 3a. Likewise, if Δ*φ_vu_* ≈ −90°, the conversion of LP-to-RHCP will be realized within the frequency band 6.45–6.85 GHz. When the conditions of *t_uu_* ≈ *t_vu_* and Δ*φ_vu_* ≈ 180° are met concurrently, one of *E_ut_* and *E_vt_* will be reversed, and then the transmission wave’s electric field will be changed to the y-direction, and it will realize the conversion of LP-to-XP within the frequency band 3.45–3.8 GHz. From the above analysis, we can conclude that the theory matches well with the simulation results.

#### 2.2.2. Reflection Performance

The reflection coefficients of the co-polarization wave are defined as *r_yy_* =|*E_yr_*|/|*E_yi_*|, where |*E_yr_*| and |*E_yi_*| are the amplitudes of the reflected and incident electric fields along the *y*-axis, respectively. Simulated reflection coefficients and phases are plotted in Figure 6. It is clear from Figure 6 that the y-polarized wave can be reflected (the reflection coefficients are less than −10 dB), and reflection phases are in the range of −90 to +90° in the range from 12.6 to 13.5 GHz, which indicates that the function of AMC can be achieved within this frequency band. Additionally, it makes the MS a strong candidate for many applications, such as in the stealth materials of radars [38] or high-gain antennas [39].

To study the influence of oblique incidence on the reflection performance, we performed simulations under different incident angles θ. The reflection coefficients and phases with varying incident angles are shown in Figure 7. It is evident from Figure 7 that the reflection phase lines are almost coincident if the oblique incidence angles are smaller than 10°; however, they will deviate for θ=10°, which indicates the low stability of the MS for oblique incidence under the reflection mode. Additionally, we will further improve oblique performance in the future.

### 2.3. Surface Current Density

The surface current distributions of the suggested MS are investigated to further clarify the physical mechanism. From Figure 3b, it is evident that the weak and the strong resonant frequencies are about 3.65 GHz and 6.42 GHz, respectively, where PCRs are high. The MS’s distributions of surface current are depicted in Figure 8 at the weak and strong resonant frequencies under the transmission mode. The distributions of surface current at the two resonant frequencies can explain the converter’s physical mechanism. It can be clearly shown from Figure 8a that, at 3.65 GHz, when the x-polarized incident EM wave impinges on the MS from the upper space, induced currents I1, I2, and I3 are excited; in the meantime, there are two pairs of anti-parallel currents on the metal layer 1 to layer 3 of the MS. One pair of anti-parallel currents is I1 and the component is I2y of I2, while the other is I3 and the component is I2x of I2, which will excite two dipolar oscillations (magnetic dipoles), namely, m1 and m2, respectively; then, the corresponding induced magnetic fields H1 and H2 are excited, respectively. As shown in Figure 8a, the magnetic fields H1 and H2 are along the x direction and the -y direction, respectively. Since H1 is parallel to the electric field component Ei of the incident EM wave, there will be a cross-coupling between H1 and Ei, resulting in the EM wave polarization conversion from x to y. While H2 is parallel to the incident magnetic field component Hi and cannot contribute to cross-polarization waves, it can increase polarization conversion by decreasing Hi in the opposite direction.

From Figure 8b, it is evident that at 6.42 GHz, when an upper-space x-polarized EM wave impinges onto the MS, the induced currents I1_1_, I1_2_, I2, and I3 are excited; meanwhile, one pair of anti-parallel currents and two pairs of parallel currents come into being on the metal layer 1 to layer 3 of the MS. The dipolar oscillation m1 is caused by the pair of anti-parallel currents I1_2_ and the component I2y of the induced current I2, and the associated induced magnetic field H1 is produced. One pair of parallel currents is the current I1_1_ and the component I2y of the current I2, while the other is the current I3 and the component I2x of the current I2, which will excite dipolar oscillations e1 and e2, respectively, and then the induced electric fields E1 and E2 are excited, respectively. As shown in Figure 8b, H1 and E2 are along the -x and x directions, respectively, whereas E1 is along the -y direction. Due to the antiparallel nature of the induced magnetic field H1 to the induced electronic field E2 and the electric field component Ei of the incident EM wave, there will be cross-coupling among the magnetic field H1, the electronic field E2, and Ei, leading to the conversion of the co-polarization wave to a cross-polarization wave. Additionally, the incident magnetic field component Hi and E1 are parallel, and they will produce cross-polarization waves. However, the two generated cross-polarization wave directions are opposite to each other, and their interactions weaken the polarization conversion; therefore, the PCR is not high at 6.42 GHz. It can be concluded that the transmissive polarization conversion feature is brought on by the magnetic dipole and the cross-coupling response.

The surface current distributions of the metal layer 3 and layer 4 are shown in Figure 9 for the reflection mode at 13 GHz. We can see from Figure 9 that at 13 GHz, when the y-polarized EM waves illuminate on the MS from the lower space, induced currents I1–I8 are excited, meanwhile, four pairs of anti-parallel currents come into being on the metal layer 3 and layer 4 of the MS. Additionally, the four pairs of anti-parallel currents are the currents I1 and I5, the currents I2 and I6, the currents I3 and I7, and the currents I4 and I8, respectively, which will excite dipolar oscillations m1, m2, m3, and m4, respectively, and then the corresponding induced magnetic fields H1–H4 are excited, which can realize the AMC function. According to the above analysis, we can conclude that the AMC function is induced by the magnetic dipole.

## 3. Fabrication and Experimental Results

To experimentally test the effectiveness of the suggested MS, we fabricated a sample made up of 26 × 26 unit cells with a size of 468 × 468 mm^2^ (9.36 × 9.36 λo at 6 GHz and 20.06 × 20.06 λo at 13 GHz). A three-layer dielectric substrate and a four-layer copper patch were fused to laminate the sample, and the metal circuits used corrosion technology from Shenzhen Hongsheng Circuit Technology Co., Ltd.; the top and bottom details are illustrated in Figure 10. The measurements were made in an anechoic chamber, and they were split into two parts. One is the transmitted mode from the upper space, as shown in Figure 11a, and the other is the reflected mode from the lower space, as shown in Figure 11b.

In order to assess the transmission coefficients and phases, we sandwiched the constructed sample between two linearly polarized horn antennas, as depicted in Figure 11a. The Prosund SP809A vector network analyzer (VNA) was connected to the two horn antennas using coaxial cables so that one of the horn antennas launched an x-polarized wave and the second horn antenna simultaneously received an x-polarized wave (co-polarized). First, the VNA was calibrated with open, short, and straight-through circuits. The transmission coefficients |Txx| and phases were measured. After that, we switched to the y-polarized horn antenna as the reception antenna (cross-polarized). The transmission coefficients |Tyx| and phases were also obtained. Simulated and measured coefficient and phase results within the frequency band 2–8 GHz are shown in Figure 12a. We find that the measured results are consistent with the simulated results, and the small discrepancies come from the installation errors of the prototype and the measurement errors of the sample, which confirms the accuracy of the design of the transmission mode.

In the reflection measurement setup, the transmitting horn antenna and the receiving horn antenna were placed side by side and connected with the Keysight E5080B vector network analyzer (VNA) using coaxial cables, which were placed 50 cm directly in front of the sample, as depicted in Figure 11b. The transmitting horn antenna (horizontal) was aligned with the receiving horn (horizontal) to measure co-polarized reflection coefficients and phases. The measurement curves are plotted in Figure 12b. It can be concluded from Figure 12b that the experimental results are consistent with simulations, and the slight variations between measurements and simulations are caused by the reflection mode’s angular instability, fabrication errors, prototype installation errors, and sample measurement errors. This indicates the design is accurate for the reflection mode.

## 4. Conclusions

In this paper, a transmission-reflection-integrated multifunctional passive MS has been proposed by elaborately designing the H-shaped chiral grating-like micro-structure and open square patches to the MS, which can transmit the x-polarized EM wave from the upper space to achieve the conversions of LP-to-LHCP, LP-to-XP and LP-to-RHCP within the frequency bands of 3.05–3.25, 3.45–3.8, and 6.45–6.85 GHz, respectively, and reflect the y-polarized EM wave from the lower space to realize the AMC function in the range from 12.6 to 13.5 GHz. Additionally, the PCR is up to −0.52 dB for the LP-to-XP conversion. Taking the transmission mode as an example, the physical mechanism of multi-functionalities was developed. Furthermore, the mechanisms were explained by studying the multifunctional MS’s current distributions in the entire space. A 26 × 26-element (468 mm × 468 mm) MS prototype was designed and measured to demonstrate its multifunctional operation. All the simulated results agree well with the experiment results, which implies the validity of the design. Our research offers a practical approach for integrating more different functionalities into one simple MS, and our suggested MS also exhibits endless potential for the wavefront control of the entire space.

## Figures and Tables

**Figure 1 materials-16-04242-f001:**
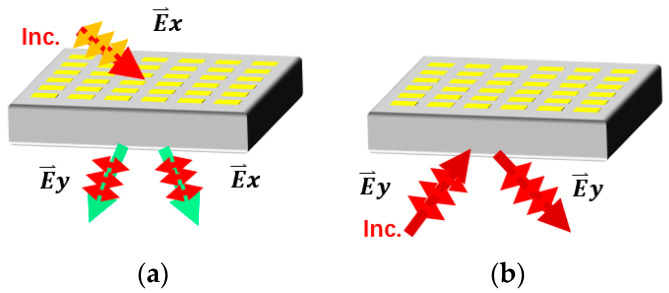
Working principle of the entire-space MS (**a**) for the transmission mode and (**b**) for the reflection mode.

**Figure 2 materials-16-04242-f002:**
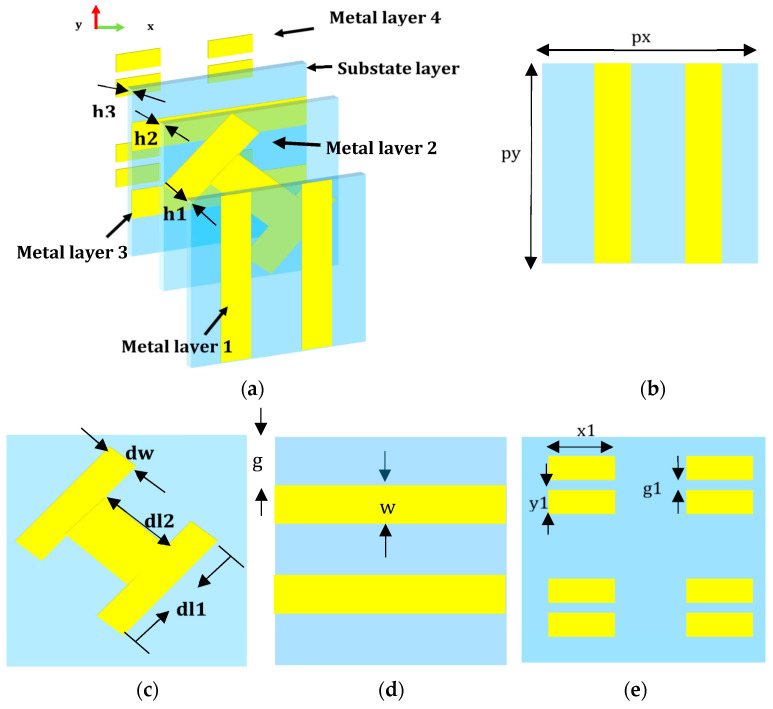
Illustration of the proposed MS unit cell. (**a**) Schematic of a unit consisting of four metal layers and three substrate layers. (**b**–**e**) Metal layer 1 to layer 4.

**Figure 3 materials-16-04242-f003:**
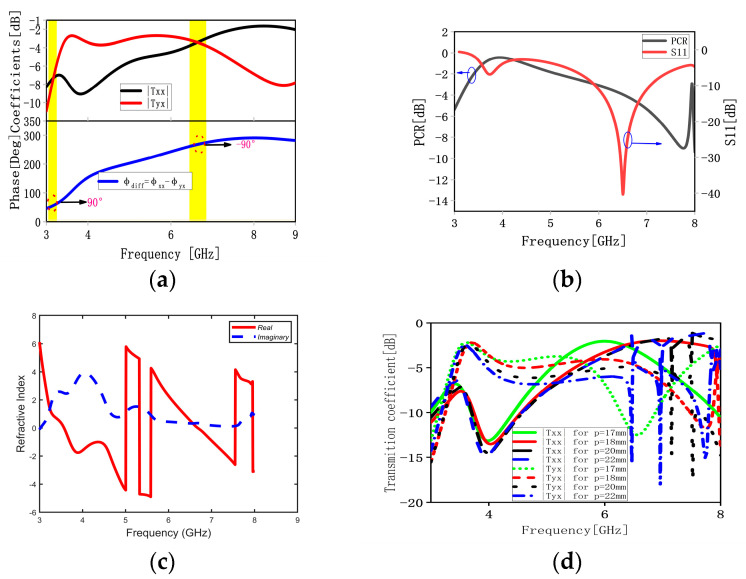
Simulated transmission coefficients, phase differences, the PCRs, reflection coefficients, and refractive indexes under the x-polarized wave from the upper space: (**a**) the transmission coefficients and the phase differences; (**b**) the PCR and S11; (**c**) the refractive indexes of the proposed MS; and (**d**) transmission performances with different periods p.

**Figure 4 materials-16-04242-f004:**
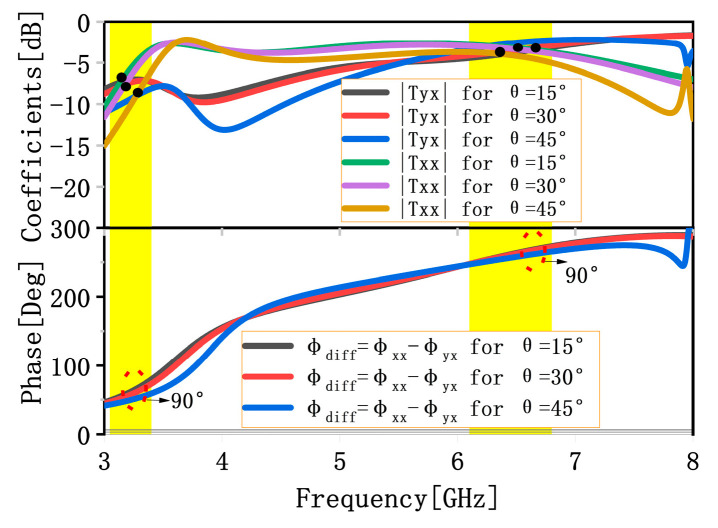
Simulated transmission coefficients and phase differences in different incident angles θ under the x-polarized wave from the upper space.

**Figure 5 materials-16-04242-f005:**
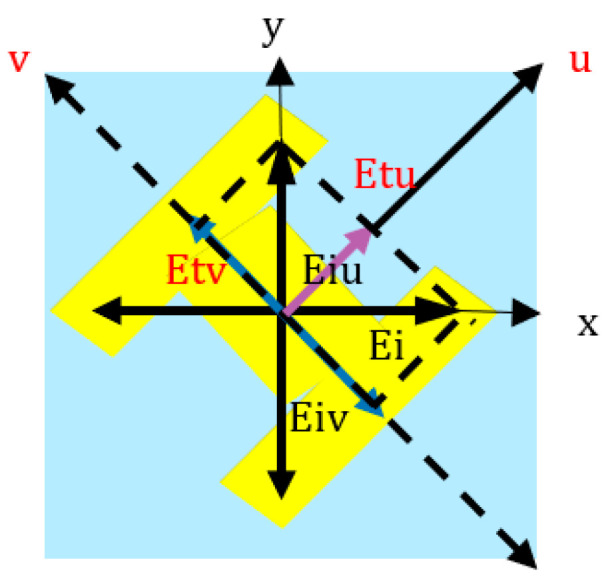
Decomposition of the electric field along the *x*-axis into the u- and v-components.

**Figure 6 materials-16-04242-f006:**
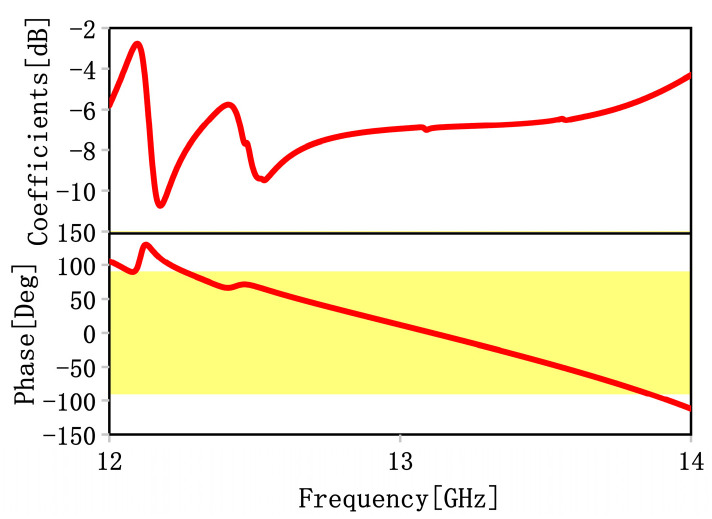
Simulated reflection coefficients and phases under the y-polarized wave from the lower space.

**Figure 7 materials-16-04242-f007:**
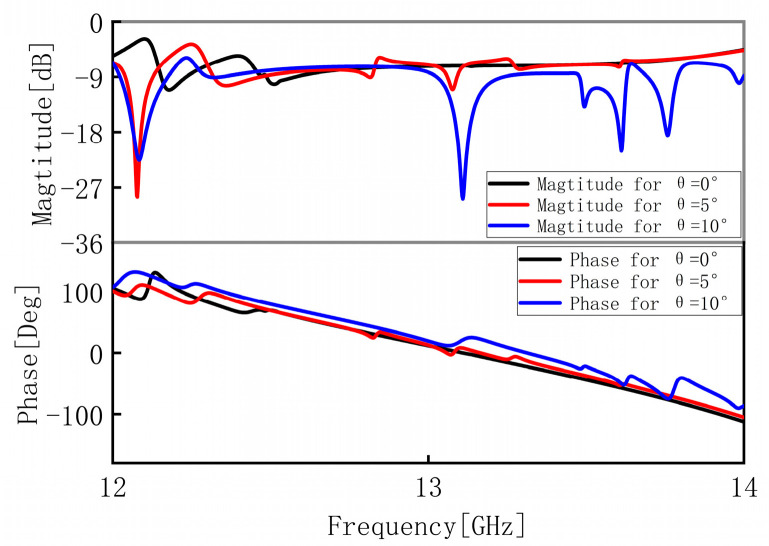
Simulated reflection coefficients and phases in different incident angles θ under the y-polarized wave from the lower space.

**Figure 8 materials-16-04242-f008:**
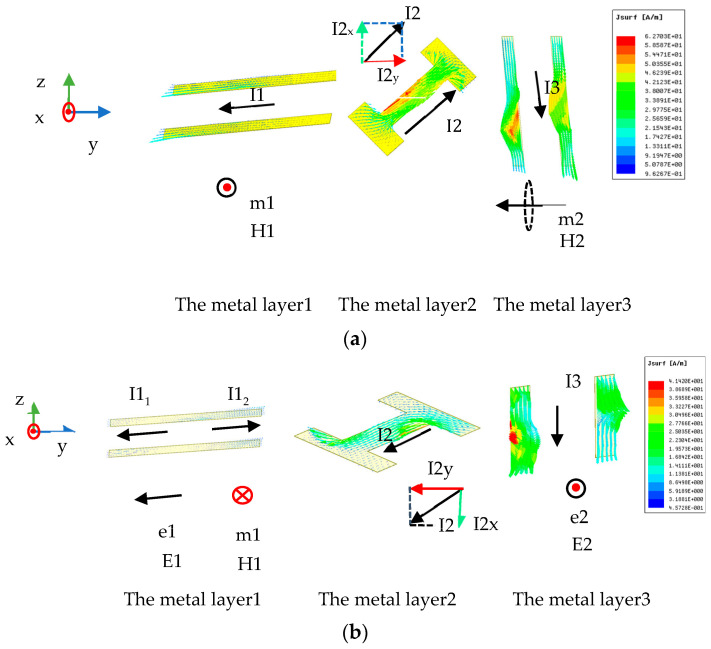
Surface current distributions of metal layer 1 to layer 3 in the transmission mode at (**a**) 3.65 GHz and (**b**) 6.42 GHz, respectively.

**Figure 9 materials-16-04242-f009:**
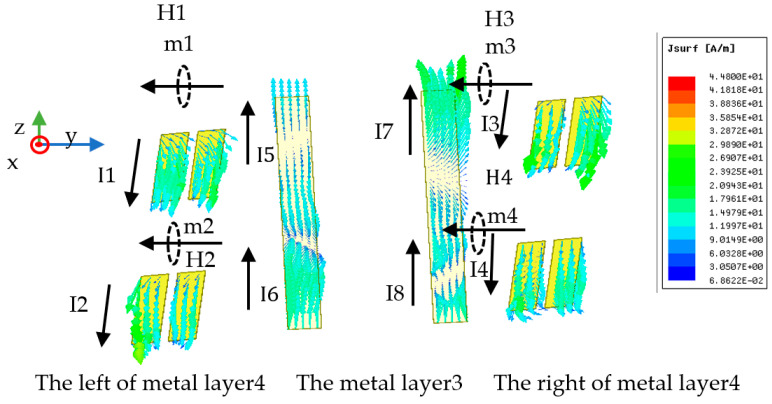
Surface current distributions of the metal layer 3 and metal layer 4 in the reflection mode at 13 GHz.

**Figure 10 materials-16-04242-f010:**
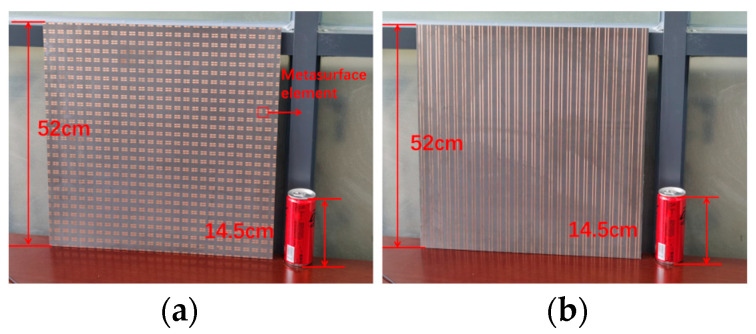
Photograph of the fabricated (**a**) top and (**b**) bottom of the MS.

**Figure 11 materials-16-04242-f011:**
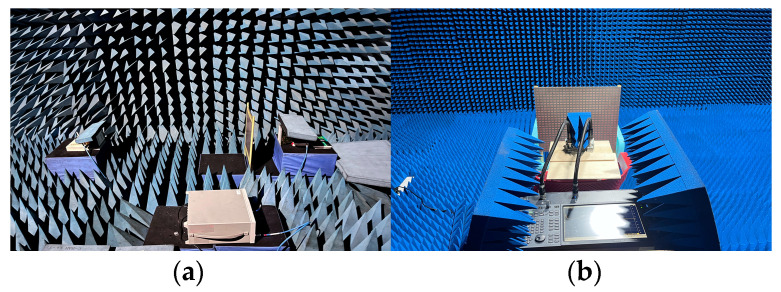
MS under test for (**a**) the transmission mode and (**b**) the reflection mode.

**Figure 12 materials-16-04242-f012:**
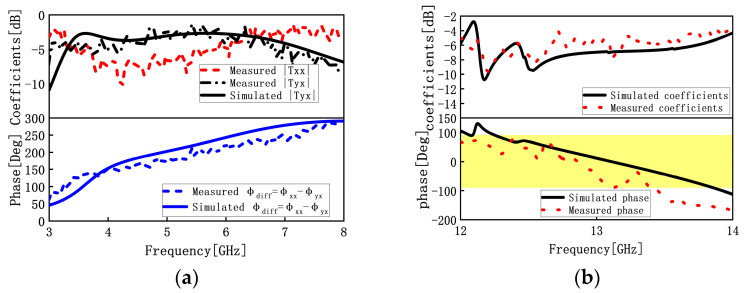
Measured results and comparisons with the simulated results (**a**) for the transmission mode and (**b**) for the reflection mode.

**Table 1 materials-16-04242-t001:** Functions within different frequency bands.

Functions	Frequency Bands (GHz)	Incident Wave Polarization	EM Wave Irradiation Spaces
LP-to-LHCP	3.05–3.25	the x-polarized wave	the upper half-space
LP-to-XP	3.45–3.8	the x-polarized wave	the upper half-space
LP-to-RHCP	6.45–6.85	the x-polarized wave	the upper half-space
AMC	12.6–13.5	the y-polarized wave	the lower half-space

## Data Availability

Not applicable.

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
