# Peer review of "Transmission-Reflection-Integrated Multifunctional Passive Metasurface for Entire-Space Electromagnetic Wave Manipulation"

_materials, 2023, doi:10.3390/ma16124242_

Round 1

Reviewer 1 Report

I thought it was an interesting paper and the theory was clear. However, I think there is a place where the explanation is insufficient. Add a description for the reader of the experiment.

1. The dimensions of the produced metasurface are difficult to understand. Include a scale bar in Figure 10 to make the size understandable to the reader. Also, please insert an arrow indicating the name and position of the device or antenna in the diagram.

2. Give details of the manufacturing method or identify the manufacturer, with the aim of allowing the reader to reproduce the measurements.

3. I didn't understand the description of Line72-94. Please explain with reference to formulas or diagrams.

4. I didn't understand why the simulation and the actual measurement are in good agreement with an antenna with poor directivity such as a horn antenna. Can this measurement ignore the influence of the antenna?

Author Response

Dear Reviewers:

Thank you for your letter and for the reviewers’ comments concerning our manuscript entitled “Transmission-reflection-integrated Multifunctional Passive Metasurface for Entire-Space Electromagnetic Wave Manipulation” (Manuscript ID: materials-2405075). Those comments are all valuable and very helpful for revising and improving our paper, as well as the important guiding significance to our researches. We have studied comments carefully and have made correction which we hope meet with approval. Revised portion are marked in baby blue in the paper. The main corrections in the paper and the responds to the reviewer’s comments are as flowing:

Reviewer #1:

  1. The dimensions of the produced metasurface are difficult to understand. Include a scale bar in Figure 10 to make the size understandable to the reader. Also, please insert an arrow indicating the name and position of the device or antenna in the diagram.

Response: We are very sorry for our negligence. We took the photos again and edited photos according to Reviewer’s comments, as shown in Figure 10.

  1. Give details of the manufacturing method or identify the manufacturer, with the aim of allowing the reader to reproduce the measurements.

Response: Considering the Reviewer’s suggestion, we have given the details of the manufacturing method or identify the manufacturer, which are marked in baby blue on lines 229-280.

  1. I didn't understand the description of Line72-94. Please explain with reference to formulas or diagrams.

Response: It is really true as Reviewer suggested. We have added a table in the description of Line72-94, as shown in Table 1.

  1. I didn't understand why the simulation and the actual measurement are in good agreement with an antenna with poor directivity such as a horn antenna. Can this measurement ignore the influence of the antenna?

Response: It is really true as Reviewer said that these measurements ignored the influence of the antenna. For the transmission mode, we subtracted the measured results without the Ms from those of the proposed MS. And for the reflection mode, we subtracted the measured results of the metal plate with the same size as the MS from those of the proposed MS. In this way, we can minimize the impact of transceiver antennas and environmental factors.

We tried our best to improve the manuscript and made some changes in the manuscript.  These changes will not influence the content and framework of the paper. And here we did not list the changes but marked in red in revised paper.

 We appreciate for Editors/Reviewers’ warm work earnestly, and hope that the correction will meet with approval.

 Once again, thank you very much for your comments and suggestions.

Kind regards,

All authors

Reviewer 2 Report

Explain the passive metasurface in the current work.

The figures have low resolution.

Discuss the role of dimensions of MSs for transmission and reflection performance.

Are the metasurfaces negative indexed?

Any measurement is performed?

Have you measured S- parameters using VNA? Explain them in detail.

What is the motivation of this work?

Moderate

Author Response

Dear Reviewers:

Thank you for your letter and for the reviewers’ comments concerning our manuscript entitled “Transmission-reflection-integrated Multifunctional Passive Metasurface for Entire-Space Electromagnetic Wave Manipulation” (Manuscript ID: materials-2405075). Those comments are all valuable and very helpful for revising and improving our paper, as well as the important guiding significance to our researches. We have studied comments carefully and have made correction which we hope meet with approval. Revised portion are marked in baby blue in the paper. The main corrections in the paper and the responds to the reviewer’s comments are as flowing:

Reviewer #2:

  1. Explain the passive metasurface in the current work.

Response: We are very sorry for our negligence. We have re-written the related contents according to Reviewer’s comments, which marked in baby blue on lines 64-67. And we have added the three references[34-36], which marked in baby blue on lines 402-407.

  1. The figures have low resolution.

Response: It is really true as Reviewer suggested that the figures have low resolution. We have edited Figure 3, 4, 6, 7, 10 and 11.

  1. Discuss the role of dimensions of MSs for transmission and reflection performance.

Response: Considering the Reviewer’s suggestion, we have added Figure 3(d) and analyzed, which marked in baby blue on lines 157-159.

  1. Are the metasurfaces negative indexed?

Response: Considering the Reviewer’s suggestion, we have simulated refractive indexes of the proposed MS based on Kramers-Kronig relationship, as shown in Figure 3(c), and analyzed, which marked in baby blue on lines 151-157. From Figure 3(c), we can find that refractive indexes are positive in some operating bands and refractive indexes are negative in some operating bands.

  1. Any measurement is performed?

Response: Yes, it is. we measured transmission coefficients and reflection coefficients for the transmission mode and reflection mode, respectively.

  1. Have you measured S- parameters using VNA? Explain them in detail.

Response: Yes, we have. Considering the Reviewer’s suggestion, we added related measurement details using VNA, which marked in baby blue on lines 288-291.

  1. What is the motivation of this work?

Response: We are very sorry for our negligence. We have added related contents, which marked in baby blue on lines 74-75.

We tried our best to improve the manuscript and made some changes in the manuscript.  These changes will not influence the content and framework of the paper. And here we did not list the changes but marked in red in revised paper.
We appreciate for Editors/Reviewers’ warm work earnestly, and hope that the correction will meet with approval.

 Once again, thank you very much for your comments and suggestions.

Kind regards,

All authors